# Versatile Aerosol Concentration Enrichment System (VACES) operating as a Cloud Condensation Nuclei (CCN) concentrator. Development and laboratory characterization.

Carmen Dameto de España[1], Gerhard Steiner[1,2], Harald Schuh[1] , Constantinos Sioutas[3], Regina Hitzenberger[1]

[1]Department of Aerosol Physics and Environmental Physics, Faculty of Physics, University of  Vienna, Vienna, 1090, Austria
[2]Institute for Ion Physics and Applied Physics, University of Innsbruck, Innsbruck, 6020, Austria
[3]Department of Civil and Environmental Engineering, University of Southern California, Los Angeles, 3620, USA

*Correspondence to*: Carmen Dameto de España (carmen.dameto@univie.ac.at)

**Abstract.** The ability of atmospheric aerosol particles to act as cloud condensation nuclei (CCN) depends on many factors, including particle size, chemical composition, and meteorological conditions. To expand our knowledge on CCN, it is essential to understand the factors leading to CCN activation. For this purpose a versatile aerosol concentrator enrichment system (VACES) has been modified to select CCN at different supersaturations. The VACES enables to sample non-volatile CCN particles without altering their chemical and physical properties. The redesigned VACES enriches CCN particles by first passing the aerosol flow to a new saturator and then to a condenser. The activated particles are concentrated by an inertial virtual impactor, and then can be returned to their original size by diffusion-drying. For the calibration, the saturator temperature was fixed at 52 °C and the condenser temperature range was altered from 5 °C to 25 °C to obtain activation curves for NaCl particles of different sizes. Critical water vapour supersaturations can be calculated using the 50 % cutpoint of these curves. Calibration results have also shown that CCN concentrations can be enriched by a factor of approx. 17, which is in agreement with the experimentally determined enrichment factor of the original VACES. The advantage of the re-designed VACES over conventional CCN counters (both static and continuous flow instruments) lies in the substantial enrichment of activated CCN which facilitates further chemical analysis.

## 1 Introduction

Atmospheric aerosols strongly influence the earth´s radiation balance directly by scattering and absorbing incoming shortwave radiation (direct effect) and indirectly by acting as cloud condensation nuclei (CCN). CCN particles can grow into cloud droplets (activate) at a critical water vapour  supersaturation (Pruppacher and Klett, 2010). Increases in droplet number concentrations for the same water content can locally increase the albedo and the persistence of clouds (Albrecht, 1989). These are respectively termed the "Twomey (first) and the cloud life time (second) aerosol indirect effects" (Twomey, 1977; Lohmann and Feichter, 2004; Charlson and Heintzenberg, 1995; IPCC, 2013). Cloud formation induced by the activation of atmospheric aerosols represents one of the main factors in determining the earth´s radiative balance (Furutani et al., 2008) and

consequently in estimating global climate change (IPCC 2013; Houghton, 2001). To constrain these uncertainties and produce more accurate predictive models of the global climate system, it is essential to improve our understanding of the activation of aerosol particles to cloud droplets and the physicochemical properties of CCN (Furutani et al., 2008).

The ability of an atmospheric aerosol particle to become a droplet depends on its size and chemical composition (Seinfeld and Pandis, 2006). Changes in aerosol size and composition during aging increases a particle's susceptibility to CCN activation (Fierce et al., 2013). Numerous studies have been made in recent years in estimating the properties that enable particles to act as CCN (Burkart et al., 2011; McFiggans et al., 2006). Particle size exerts the strongest influence on the ability of particles to act as CCN, since soluble mass changes with the third power of particle diameter (Andreae and Rosenfeld, 2008; Ervens et al., 2007). Chemical effects, however, can modify the ability of a particle to act as a CCN (Kreidenweis et al., 2006; Laaksonen et al., 1998). Insoluble but wettable particles can promote droplet formation. Hydrophilic substances strongly facilitate droplet activation whereas hydrophobic substances can inhibit droplet formation (Andreae and Rosenfeld, 2008). According to Köhler theory, which describes the effects involved in cloud droplet activation, the critical saturation $S_c$ is a function of a particle's chemical composition and size. This includes the number of potential solute molecules and their solubility (Tomasi, 2017). Compared to most soluble organic compounds, inorganic salts have a higher solubility which results in a lower critical supersaturation for particles of equal size. Hings et al. (2008) showed that the $S_c$ of 100 nm ammonium sulphate and adipic acid particles is 0.15 % and 0.27 % respectively. The presence of slightly soluble aerosol material will further decrease the $S_c$ (Kulmala et al., 1997; Zhang et al., 2012; Nenes et al., 2002). Organic compounds in particular can influence CCN activity by several mechanisms: contribution to solute matter, reduction of surface tension (e.g. Hitzenberger et al., 2002) and formation of hydrophobic surface films (Dusek et al., 2006; Kanakidou et al., 2005). The effects of organic compounds on CCN activation are still not well known. The large variety of atmospheric organic molecules and the numerous types of inorganic and organic aerosol material which are internally or externally mixed (e.g. Okada and Hitzenberger, 2001) further increase the difficulties in understanding CCN activity in the atmosphere. To gain more knowledge on atmospheric CCN, more characterisation studies of CCN are needed. Numerous laboratory experiments have been carried out to explore the CCN activity of particles consisting of relatively simple model chemical species such as $(NH_4)_2SO_4$, NaCl and organic substances commonly detected in the atmosphere (e.g. Furutani et al., 2008; Giebl et al., 2002; Henning et al., 2005; Hings et al., 2008; Rose et al., 2008). As the number concentrations of CCN are quite low (mostly up to some 100 /cm³), chemical analyses of activated CCN are quite challenging. To facilitate exploration of CCN properties we therefore introduce an adapted concentration enrichment method to increase CCN concentrations for further analysis.

The versatile aerosol concentration enrichment system (VACES) consists of an ambient particle concentrator developed by Sioutas (1999). A VACES achieves enrichment of fine and ultrafine particles by first growing them to super-micron droplets in a supersaturation/ condensation system, and then concentrating them via a virtual impactor (Geller et al., 2005; Kim et al., 2001a; Wang et al., 2013a). These concentrated particles are enriched without being altered chemically or physically, except for the condensed water vapour, which can subsequently be removed by diffusion drying. In case of particles containing semivolatile or volatile species, some chemical changes may occur in the temperature range used in the VACES. The VACES

has been used in a number of human and animal toxicological studies of the adverse effects of exposure to particulate matter (e.g. Klocke et al., 2017; Ljubimova et al., 2018; Wang et al., 2018)

The VACES has been also used to measure particulate matter in different locations such as in an underground railway station (Loxham et al., 2013), in Mexico City (De Vizcaya-Ruiz et al., 2006), in California´s San Joaquin valley (Plummer et al., 2012) and in the Netherlands (Steenhof et al., 2011). Studies on emissions and exhaust particles from cars and their health effects were conducted with a VACES system by Tzamkiozis et al. (2010), Gerlofs-Nijland et al. (2010). Other applications of VACES included the determination of chemical properties of particulate matter sampled on filters (Verma et al., 2011; Wang et al., 2013b). Modifications and further developments of VACES for different applications were performed by e.g. Geller et al. (2005), Pakbin et al. (2011), Saarikoski et al. (2014) and Wang et al. (2013a). Several recent studies corroborated the integrity of the VACES as a particle concentrator (e.g. Zhao et al., 2005; Ning et al., 2006; Ntziachristos et al., 2007 and Saarikoski et al., 2014)

Up till now, VACES were operated at quite high water vapour supersaturations ratios (typically on the order of 1.5-1.8) to enrich ultrafine particles with sizes down to 10 nm efficiently. These high supersaturations ratios, however, are far too high to study CCN activation. In the atmosphere, typical supersaturations are much lower (around or below 0.5%; Reade et al., 2006 and Aalto et al., 2000). For atmospheric CCN activation the chemical composition of particles is relevant in a limited size range between 30 nm and 200 nm (McFiggans et al., 2006). Particles smaller than 30 nm will hardly activate and particles larger than 200 nm are sufficiently large to activate practically irrespective of chemical composition.

In our current study, we modified a VACES to operate at low supersaturations to enable enrichment of CCN. For the calibration runs, we used NaCl particles and Polystyrene Latex (PSL) particles with sizes in the range of 30 nm to 200 nm, i.e. the range where the chemical composition is relevant for particle activation at typical atmospheric supersaturations. We demonstrate the stability and reproducibility of this modified VACES for supersaturations < 0.6 %, which are far lower than the supersaturations used in the conventional VACES setup. As an additional aspect, the concentration enrichment factor of the VACES' virtual impactors is measured in a novel way, i.e. from activation curves.

## 2 Theoretical background

Classical Köhler theory describes the equilibrium size of a droplet containing soluble material as a function of the water saturation ratio S and combines the Raoult and Kelvin effects. The Köhler equation is expressed by Eq (1).

$$S = \frac{p}{p_0} = \left(1 + \frac{6\, i\, m_s\, M_w}{M_s\, \rho_w\, \pi\, d_d^3}\right)^{-1} exp\left(\frac{4\, \sigma_w\, M_w}{\rho_w\, R\, T\, d_d}\right) \tag{1}$$

where S is defined as the actual vapour pressure $p$ divided by the saturation vapour pressure $p_0$, $i$ the van´t Hoff factor, $m_s$ the mass of the solute, $M_w$ and $M_s$ the molecular masses of water and solute respectively, $R$ the universal gas constant, $T$ the temperature in K, and $\rho_w$ and $\sigma_w$ the density and the surface tension of water. $d_d$ is the droplet size. This Köhler equation is difficult to solve analytically. For small supersaturations (i.e. S slightly above 1), a Taylor series expansion truncated after the

first term can be used to obtain the critical supersaturation $S_c$ ($S_c = S-1$) and the critical droplet size $d_{d,c}$ as (Seinfeld and Pandis, 2006):

$$S_C = \frac{p}{p_0} - 1 = \sqrt{\frac{256\,\sigma_W{}^3\,M_W{}^2\,M_S}{27\,i\,R^3\,T^3\rho_W{}^2\,\rho_S\,d_p{}^3}} \tag{2}$$

$$d_{d,c} = \sqrt{\frac{18\,R\,T\,i\,m_S}{4\,\pi\,\sigma\,M_S}} \tag{3}$$

The activation and growth to droplet sizes of insoluble wettable particles is governed by the Kelvin effect (Hinds, 1999). The critical supersaturation for a spherical particle with a diameter $d_p$ is given by the Kelvin equation:

$$S_c = \frac{p}{p_0} - 1 = exp\left(\frac{4\,\sigma_w\,M_w}{\rho_w\,R\,T\,d_p}\right) - 1 \tag{4}$$

In this study, we use the parameters given in Table 1 to calculate critical supersaturations.

Critical supersaturations for particles of a given size and chemical composition can be obtained from measured activation curves, where the activation ratio (defined as the ratio of the number concentrations of activated and unactivated particles) is plotted as a function of supersaturation. The 50 % point of this curve then gives the measured critical supersaturation for these particles. In the CCN-VACES, activation curves are obtained in a slightly different way by measuring the activation ratio as a function of the exit temperature $T_{out}$ (see below). Knowing the temperature where 50 % of the particles activated and the

particle size, the critical supersaturation can be calculated from Eq. (2).

**3 Instrumentation**

The new CCN-VACES is based on the previously developed VACES system (Kim et al., 2001a, 2001b) and has numerous new features and modifications. The original VACES has two condenser tubes. In our setup, both tubes are operated as prescribed, the aerosol flow passes both tubes, but the actual measurements are performed only at one of the tubes. The VACES operation

principle has not been modified. This consists on first saturating the aerosol flow with water vapour and then cooling it down in the condenser to achieve supersaturation, particle activation and growth by water condensation. The grown particles are concentrated in the minor flow of the virtual impactor, while particles smaller than the cut size of the virtual impactor are removed from the system in the major flow. From the original VACES, only the virtual impactors and the temperature regulator/ chiller for the condenser tubes were used, while the saturator and condenser parts were changed (see below) in the

CCN-VACES.

**3.1 Saturator**

A new vertical heat-insulated stainless steel cylinder tank 40 cm long and 40 cm in diameter, partially filled with ultrapure water (Direct-Q5®, Millipore, Billerica, MA) is placed on two 18 cm diameter electric hotplates (1500 W, EKP3582, Claronc®), which are switched on and off with a solid-state relay controlled by an Arduino microcontroller board

(ARDUINO®; an open-source platform which consists of both hardware and software to be used for process control) to keep the water temperature constant within 0.2 °C. A small water pump inside the tank keeps the water well mixed and ensures a homogeneous temperature profile within the water tank. Temperature sensors measure the water temperature $T_w$ and the air temperature inside the saturator vessel directly before the aerosol outlet (saturator temperature $T_s$). Aerosol is introduced into the space above the heated water where it is saturated with water vapour and subsequently led into the cooling tubes of the VACES. In these cooling tubes, the water vapour becomes supersaturated and condenses on the particles. The total flow rate of the system is determined by the flow rate of the virtual impactors (105 L/min each), so the test aerosol used here (monodispersed NaCl aerosol produced with a DMA with an aerosol flow rate of 5 L/min see below) had to be diluted with ambient filtered air to achieve the necessary volume flow. The connecting tube between the saturator and the condenser is equipped with an isolating mantle enclosing a heating wire wrapped around this tube to avoid temperature changes and premature water condensation.

## 3.2 Condenser

The water vapour saturated air is drawn into two parallel 1 m long steel tubes consisting each of a 2.2 cm diameter inner tube surrounded by a 7.62 cm diameter outer tube. The space between the two concentric tubes is filled with an ethylene glycol/ water (1:1 by volume) coolant. The outer tubes are connected at the beginning and at the end of the tubes with a commercially available recirculating chiller (Thermocube 300 1D 1 LT Solid State Cooling Systems, Pleasant Valley, NY), which regulates the coolant temperature within +/- 0.1 °C. The coolant temperature in the tube is defined here as condenser temperature $T_c$ . In order to preserve the pre-set temperature, both condenser tubes are insulated with polyethylene foam pipes. In these condenser tubes the warm saturated air exiting the saturator is cooled below the dewpoint to achieve supersaturation, so aerosol particles can activate and grow to droplets with sizes about 2.5 – 3 µm (Daher et al., 2011; Kim et al., 2001a). At the exit of the condenser, the air temperature $T_{out}$ is measured.

## 3.3 Virtual impactor

The two condenser tubes are followed each by a virtual impactor with a nominal cut point of 1.5 µm. Virtual impactors classify particles according to their inertia by separating them into two streams according to their aerodynamic diameters (e.g. Marple and Chien, 1980). The aerosol passes first through an accelerating nozzle and is directed to a collection probe where the size classification occurs. At this point a major part of the flow is diverted 90° away from the collection probe. Particles with smaller aerodynamic diameter and lower inertia follow the streamlines of the major flow, while particles with aerodynamic diameters larger than the cut-size continue moving axially in their forward path penetrating further into the collection probe with the minor flow (Kulkarni et al., 2011). The separation efficiency curve is determined by the ratio of the major and minor flows and the physical dimensions of the nozzle and collection probe.

Besides the particle-size separation the virtual impactor concentrates particles larger than its cut-size in the minor flow. The theoretical enrichment factor is equal to the ratio of the total flow rate to the minor flow rate. One characteristic of a virtual

impactor is that particles smaller than the cut-size of the impactor remain in both the major and minor flows. As the minor flow in our case is 5 % of the total flow, only 5 % of the small particles will remain with the minor flow. The construction of a virtual impactor has a strong influence on the shape of the separation efficiency curve and on particle losses (Marple and Chien 1980). The characterization of virtual impactors and their collection efficiency curves has been intensively studied by Sioutas (1994, 1996, 1999). The virtual impactors used in our study are identical to those used by Kim et al. (2001a) with a major flow of 105 L/min and a minor flow of 5 L/min.

In our setup, the minor flow exiting the virtual impactor is passed through a diffusion dryer and then split into two flows. One flow (0.6 L/min) is led to a condensation particle counter, CPC (Grimm 5412, flow rate 0.6 L/min), while the other flow (4.4 L/min, regulated with a flow controller) is vented to the exhaust.

The collection efficiency curves and particle losses of the virtual impactor used here were calculated by Sioutas (1994). The theoretical concentration enrichment factor $EF_{Theo}$ is given by Eq. (5)

$$EF_{Theo} = \frac{Q_{tot}}{q_{min}} (1 - WL)\eta_{vi} \qquad (5)$$

where $Q_{tot}$ and $q_{min}$ are the total and minor flows of the impactor, respectively, and $WL$ and $\eta_{vi}$ the fractional losses and the collection efficiency (Sioutas et al., 1999). For a VACES, small values of $WL$ and the assumption that the virtual impactor collection efficiency $\eta_{vi}$ at 50 % droplet activation is around 1, the enrichment factor approaches the theoretical enrichment factor given by Eq. (6)

$$EF_{Theo} = \frac{Q_{tot}}{q_{min}} \qquad (6)$$

### 3.4 Monodisperse aerosol generation

NaCl particles were generated by nebulizing a 10 g/L solution of sodium chloride (Applichem GmbH, > 99 %) in ultrapure water with a Collison atomizer (TSI, 3076) operated with particle free air at 1.2 bar producing a 2 L/min flow. This flow was diluted with 3 L/min of dry clean air and dried to relative humidities < 15 % with a diffusion dryer. The 5 L/min total flow was neutralized with a soft X-ray charger (TSI, 3087) to bring the aerosol into charge equilibrium. A Vienna type Differential Mobility Particle Sizer (DMPS) (Winklmayr et al., 1991) was operated in a closed loop arrangement and used to generate monodisperse particles with sizes of 30 nm, 100 nm, 150 nm and 200 nm mobility equivalent diameter. The size selective behaviour of the DMA was tested with monodisperse PSL particles (Polyscience Inc. Warrington, PA) of various sizes. Before each measurement run all flow rates were checked with a Gilibrator High flow generator cell (Sensidyne™, 800265). PSL particles with sizes of 100 nm and 150 nm were used to check the CCN-VACES calibration for insoluble particles.

### 3.5 Experimental set-up

The CCN-VACES was evaluated measuring the enriched particle number concentration at different temperature settings of the saturator and condenser.  For these measurements, monodisperse NaCl aerosol was drawn through the saturated water vapour atmosphere in the heated water tank before entering the cooling sections (condenser tubes) of the VACES. In order to have an

incoming particle number concentration reference, the particle number concentration was measured before entering the saturator tank with the first condensation particle counter (CPC, Grimm 5412; flow rate 0.6 L/min), which corresponds to CPC1 in Fig. (1). Tank water temperature, $T_w$ was set at 52 °C and kept constant within 0.2 °C. Before starting the measurements the water level in the tank was checked to ensure a water volume of 24 L. Water heating time was around one hour followed by a stabilisation time of 30 minutes to achieve a constant water temperature. As the system is very sensitive to temperature variations, keeping the water temperature constant was very important. The aerosol residence time in the tank was about 8s to ensure saturation of the aerosol stream. Although the saturated aerosol is drawn through both tubes in parallel, only one tube was used in the following experiments. Prior to entering the condenser tube the inlet particle number concentration was measured with an isokinetic sampling tube (8mm diameter)  connected to a diffusion dryer to remove excess humidity followed by  a second CPC (Grimm 5412; flow rate 0.6 L/min) referred to as CPC 2 in Fig. (1). This number concentration is defined as the inlet concentration $C_{in}$. After passing through the condenser tube the aerosol is separated into two size classes by the virtual impactor (nominal cut size 1.5 µm aerodynamic diameter). The minor flow from the virtual impactor containing the activated droplets is connected to a diffusion dryer to dry the droplets to their original size. A third CPC (Grimm 5412; flow rate 0.6 L/min; referred to as CPC 3 in Fig. (1)) was used to measure the number concentration of the enriched particles.

Before each calibration run, the system was checked for leaks and all flow rates were measured. The output concentration of the atomizer was checked for fluctuations, as these fluctuations are indicators of leaks or impurities in the system. Measurements were started after $T_w$ was found to be stable (see above). For the calibration curve measurements, the aerosol was passed through a DMA to select a particle size, led through the saturator tank and subsequently into the condenser which was set to a fixed temperature. At the start of a calibration run, $T_c$ was set at 30 °C followed by 25 °C. At these two temperatures no noticeable changes in the condenser outlet concentration were observed. $T_c$ was subsequently reduced in 1°C increments. At each temperature setting, ca. 20 minutes were necessary for the system to stabilize. Inlet concentration, outlet concentration, $T_w$, $T_s$, $T_c$, $T_{out}$ and laboratory temperature were recorded after stabilization for 5 min with a time resolution of 1 sec to ensure good statistical results. Each data point in Figures 3 - 6 therefore corresponds to the mean value of 300 individual measurement points. The time required for an activation curve measurement was 6 to 8 hours.

## 4   Results and discussion

### 4.1 Measurement of the enrichment factor of the CCN-VACES set-up

The CCN-VACES was tested first by determining the experimental concentration enrichment factor using NaCl and PSL particles of different sizes. PSL particles were used to check whether the CCN-VACES operates well also for non-hygroscopic aerosols. For the measurements $T_w$ was set at 52 °C (+/- 0.2 °C) and $T_c$ was decreased  from 20 °C to 5 °C with steps of 1 °C (+/- 0.1 °C).  The concentration of the minor flow of the VI is enriched by a factor ideally equal to the ratio of total sample flow rate to the minor flow rate (Kim et al., 2001). In order to determine the experimental enrichment factor of our modified setup, activation curves were used in which the ratio of the outlet concentration to the inlet concentration is plotted over the

temperature difference $\Delta T$ between the saturator temperature $T_s$ and the condenser temperature $T_c$. In the measurements of enrichment factors, $\Delta T$ was used as independent variable in order to make our results comparable to previously published enrichment factors, which investigated the enrichment factor in terms of $T_c$. The experimental enrichment factor curves obtained for 100 nm NaCl and PSL particles are shown in Fig (2).

The maximum experimental enrichment factor obtained for different particle sizes is given in Table 2. These experimental enrichment factors ($EF_{exp}$) can be compared with the theoretical enrichment factor ($EF_{Theo}$) calculated according to Eq (6). A total flow of 105 L/min and a minor flow of 5 L/min result in an $EF_{Theo} = 21$. The experimental enrichment factor corresponds to around 80 % of the theoretical enrichment factor which corroborates the efficiency of 80 % of the original VACES setup (Kim et al., 2001b). These results can be compared with other already published studies. In the study by Geller et al. (2005) the enrichment factor for 50 nm and 170 nm PSL particles was found to be 15 and 15.3 respectively. The theoretical enrichment factor in the study by Geller et al. (2005) is 20, so their experimental enrichment factor corresponds to 75 % of the theoretical enrichment factor. These results agree with our experimental enrichment factor results.

## 4.2 Size dependence of the activation curves

The CCN-VACES was calibrated using the set-up given in Fig. (1) from activation curves obtained for monodispersed particles of known chemical composition (e.g. Rose et al., 2008; Dusek et al., 2006). One important issue in this study was to analyse the dependence of the activation of particles of different size on the temperature settings. For this experiment, $T_w$ was again set at 52°C (+/- 0.2 °C) and the condenser temperature $T_c$ was again decreased from 20 °C to 5 °C (+/- 0.1 °C) in steps of 1 °C. The actual temperature at the exit of the condenser ($T_{out}$) determines the supersaturation experienced by the particles. This temperature, however, cannot be set, so in this part of the study, we set $T_c$ and used the particles of different sizes as "supersaturation sensors" here. Different activation curves were measured for particles of different sizes. Normally, the activation ratio of an aerosol is defined as the concentration of the activated particles divided by the total particle concentration. In our case, the enrichment factor of the virtual impactor has to be taken into account, so here (Fig. 3) the activation ratio is given as $(C_{out}/C_{in})/(C_{out}/C_{in})_{max}$ where $(C_{out}/C_{in})_{max}$ is essentially the measured enrichment factor given in Table 2, so the activation curves are normalized to 1 at 100 % activation. Fig. (3) shows the normalized activation curves for particles of different sizes as a function of the temperature difference between saturator $T_s$ and condenser $T_c$ ($\Delta T$). The error bars corresponding to the standard error of the mean are included. Based on Köhler theory, the critical supersaturation is a function of $T^{-3/2}$ and $d_p^{-3/2}$ which translates into higher critical supersaturations for small particle activation compared to activation of larger particles. These higher critical supersaturations are achieved by larger $\Delta T$ or, at fixed saturator temperatures $T_s$, lower condenser temperatures $T_c$. Larger particles activate at lower critical supersaturations and, in our case, smaller $\Delta T$ (i.e. higher $T_c$). This behaviour can be observed in Fig. (3). The activation curve for 30 nm particles is shifted to the right (larger $\Delta T$) compared to the curve for 200 nm particles.

## 4.3 Reproducibility

The reproducibility of activation curves measured with the CCN-VACES was also tested. Different calibration curves were measured on different days using the same temperature settings. By comparing results from the same particle size and temperature settings, we obtained identical activation curves, which demonstrates that the CCN-VACES is reliable as a particle activator. Fig. (4) shows an example of activation curves obtained for 100 nm NaCl particles on different days, but at the same laboratory temperature $T_{lab}$= 23 °C. The data points coincide on the same fitted curve.

Further results also demonstrate that the activation curves do not depend on the inlet concentration. Curves obtained for monodisperse particles with 6000 /cm³ concentration (the highest concentration achievable in our setup) were found to be identical to curves obtained for particles of the same size and concentration of 2000 /cm³. This finding agrees with previous studies by Sioutas et al. (1996).

### 4.3.1 Dependence on ambient temperature

When measuring the activation curves, the CCN-VACES system was found to be extremely sensitive to slight variations in the temperature settings. In addition, ambient (laboratory) temperature was found to have a noticeable effect on the saturator temperature $T_s$. In order to obtain identical activation curves for particles of a certain size, also the ambient temperature has to remain fairly constant. As the CCN-VACES draws 200 L/min filtered ambient air, the temperature of the aerosol stream inside the system and consequently $T_s$ is influenced by the ambient temperature, which has an effect on the supersaturation achieved in the condenser. This behaviour was seen by comparing activation curves as function of ΔT obtained in different seasons, with different laboratory temperatures (summer: around 27 °C, winter: 22 °C). As there is no way to set the temperature of the aerosol stream at the outlet of the saturator (i.e. $T_s$), the only temperature that can be controlled in the saturator is the water temperature $T_w$. For the experiments shown in Fig. (5), the water temperature was always set to $T_w$= 52 °C. Mixing of the aerosol stream with filtered ambient air resulted in a saturator temperature $T_s$= 39 °C in summer and $T_s$= 37.5 °C in winter. $T_s$ has a strong influence on the resultant ΔT and consequently on the actual supersaturation experienced by the particles. As a result, the activation curves measured during summer are shifted to the left (red curve in Fig. 5) compared to the activation curves measured in winter (blue curve) for the same setting of ΔT.

### 4.4 Determination of the supersaturation in the CCN-VACES

In order to determine the supersaturation experienced by the particles downstream of the condenser tube, a temperature sensor was inserted in the connector between the condenser and the virtual impactor. As the flow is turbulent, the temperature sensor was put at the inner surface of the (thermally insulated) connector. This temperature is referred to here as $T_{out}$ and corresponds to the temperature of the particles right before they enter the virtual impactor. For the calculation of the actual supersaturations achieved in the condenser, activation curves are plotted as a function of $T_{out}$. $T_{out}$ at the 50 % point of the curves was used to obtain $S_c$ from Eq. (2) for NaCl particles and Eq. (4) for PSL particles. In the following Fig. (6), activation curves of NaCl

particles are shown with the corresponding error bars. In Fig. (6) the activation curves have a smooth slope which agrees to results obtained in other studies such as the one by Giebl et al. (2002) or Frank et al. (2007) .

For the determination of the critical supersaturation, a sigmoidal curve was fitted to each data set. As the supersaturation is proportional to $T^{-3/2}$, smaller particles activate at higher supersaturations and therefore at lower temperature $T_{out}$. The curve for 200 nm particles shows that 30 % of the particles are already activated at $T_{out}$ = 31 °C. For 100 nm particles, this fraction is 10%. In contrast to Fig. (3) where the activation curve for 30 nm particles is shifted to the right, in Fig. (6) the activation curve for 30 nm particles is shifted to the left. This is due to the different presentation: in Fig. (3) the activation ratio is plotted as a function of ΔT, and in Fig. (6) as a function of $T_{out}$.

The curves also show that activation is extremely sensitive to $T_{out}$. The difference in $T_{out}$ from one activation curve to the next in Fig. 7 is only 0.2 °C. For 30 nm particles, $T_{out}$ at 50 % activation is 28.6 °C whereas for 100 nm NaCl particles it is 28.8 °C. As shown in Figures 7a – 7d, the temperature at which 50 % of the particles activate can be obtained. Based on this temperature the supersaturation can be determined from Eq. (2) using the values of the parameters given in Table 1. Results of the corresponding temperature together with the calculated supersaturation are given in Table 3. The critical supersaturation for PSL particles was calculated from Kelvin theory (Eq. (4)). As PSL particles are insoluble the critical supersaturation only depends on the Kelvin term previously defined in the section 2. The results presented in Table 4 show that PSL particles need higher critical supersaturations to activate than NaCl particles of comparable size. For example 100 nm PSL particles need a critical supersaturation of 2.1% whereas 100 nm NaCl particles activate at 0.1% critical supersaturation. These results are in agreement with Köhler theory: the more hygroscopic the particles the lower the supersaturation needed to activate.

**5 Summary and Conclusions**

In this work we have shown that the overall performance of the CCN-VACES is similar to that of the original VACES. Our experimentally determined enrichment factor was ca. 16.5 - 17.0, which is lower than the theoretical enrichment factor of 21, but similar to experimental values obtained in other studies for the original VACES (e.g. Wang et al., 2016).

Our results show that the CCN-VACES is reliable as a particle activator and can enrich CCN at low supersaturations (<0.6%). The system is, however, dependent on ambient conditions and very sensitive to the temperature settings. To measure activation curves for particles of different sizes, the temperature settings have to remain as stable as possible. The variations of water temperature $T_w$ and saturator temperature $T_s$ during a measurement series should be less than 1°C. In this study, we used the temperature at the exit of the condenser (i.e. $T_{out}$) where 50 % of the particles activate and the known dry particle size to calculate the supersaturation. We observed that for 100nm NaCl particles, this temperature was 28.77 °C, while for 30 nm NaCl particles it was 28.56 °C. The difference between these two temperatures is only 0.21 °C whereas the corresponding particle size difference is 70 nm. If the CCN-VACES is used as a CCN concentrator, calibration runs should be performed for each supersaturation setting with test particles of sufficiently different size.

Notwithstanding the strong temperature dependence, we found that the CCN VACES is a reliable instrument to activate CCN and enrich CCN concentrations at low supersaturations, provided that the temperature settings are carefully controlled. Some changes of particles containing volatile or semi-volatile substances, however, might occur in the temperature range used in the saturator and condenser. In contrast to continuous flow (Rose et al., 2008) and static thermal gradient CCNCs (Giebl et al., 2002), the CCN-VACES provides enriched CCN concentrations for further analysis.

## Author contribution

CD: design and development of the modified VACES; experiments, data analysis, MS preparation.

GS: design of the experiment, preparation of the experimental setup, data interpretation.

HS: design and technical development of the experiment.

CS: developed the VACES system and provided technical support for the preparation of the manuscript.

RH: initiator and supervisor of this research work, advice with experiment and data analysis, extensive input to MS text.

## Acknowledgements

The authors wish to thank Anna Wonaschütz and Christoph Hitzenberger for helpful discussions and Christian Tauber for the data acquisition programme for the simultaneous recording of particle concentrations and temperature settings.

## Competing interests

The authors declare that they have no conflict of interest.

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

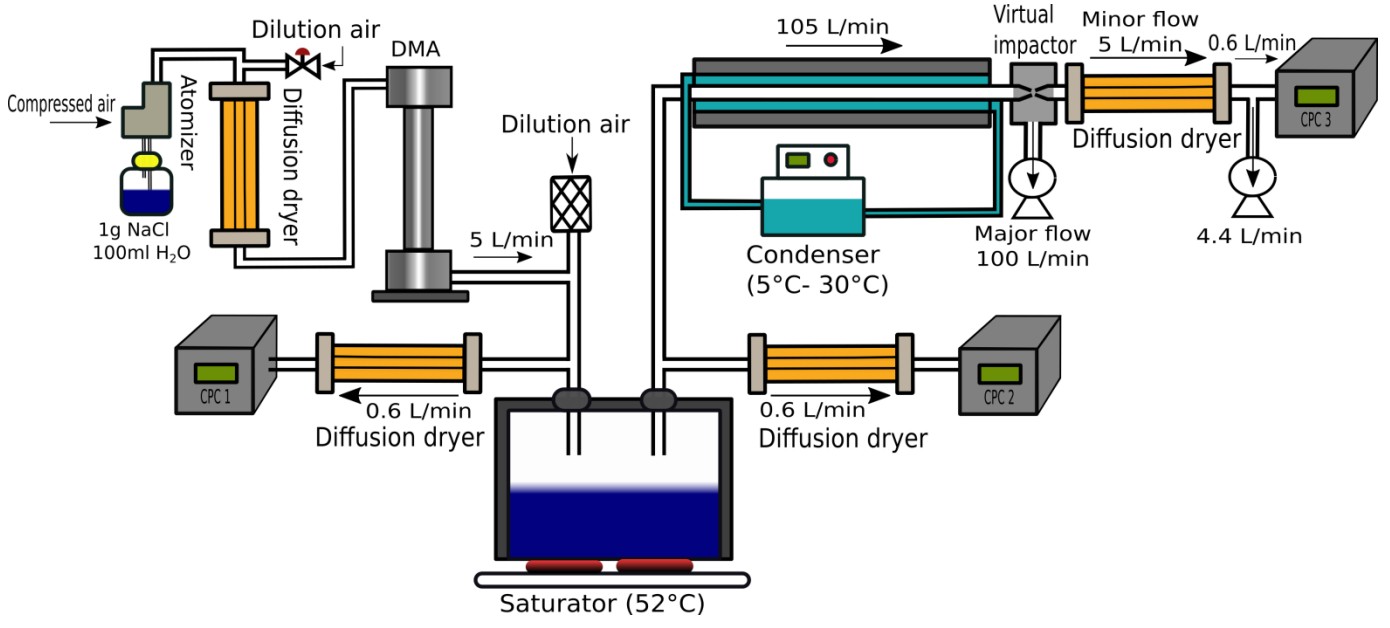

**Figure 1:** Set-up for the CCN-VACES calibration. Only one condenser tube is used for the calibration.

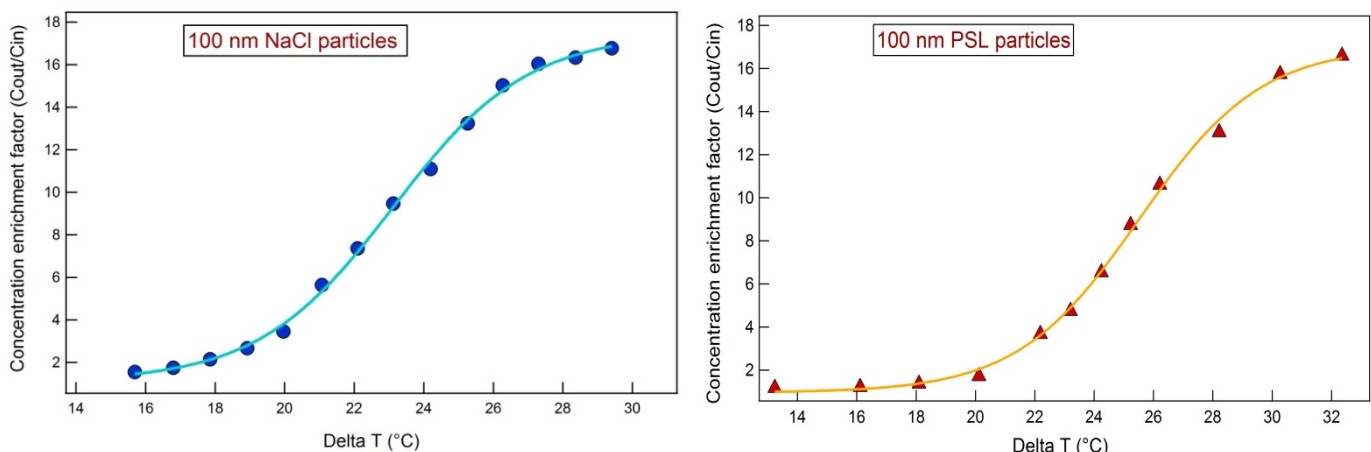

**Figure 2: a)** Experimentally determined concentration enrichment factor for 100nm NaCl particles at different ΔT (Difference between the saturator temperature $T_s$ and the condenser temperature $T_c$). **b)** Experimentally determined concentration enrichment factor for 100 nm PSL particles as a function of ΔT. The maimum experimental enrichment factors correspond to 100% activation.

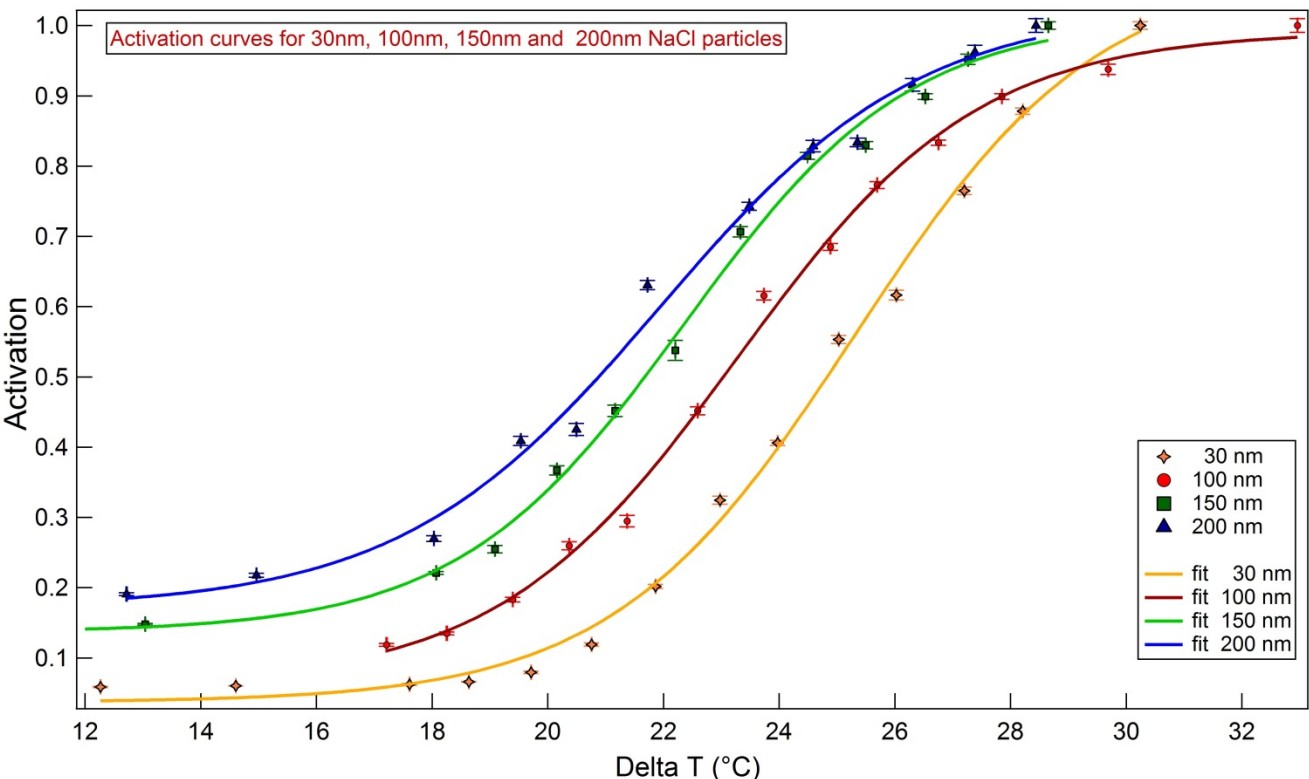

**Figure 3:** Activation curves for NaCl particles of different size at different ΔT, which is the temperature difference between the saturator temperature $T_s$ and condenser temperature $T_c$ . The measured values (points) are fitted with sigmoidal curves.

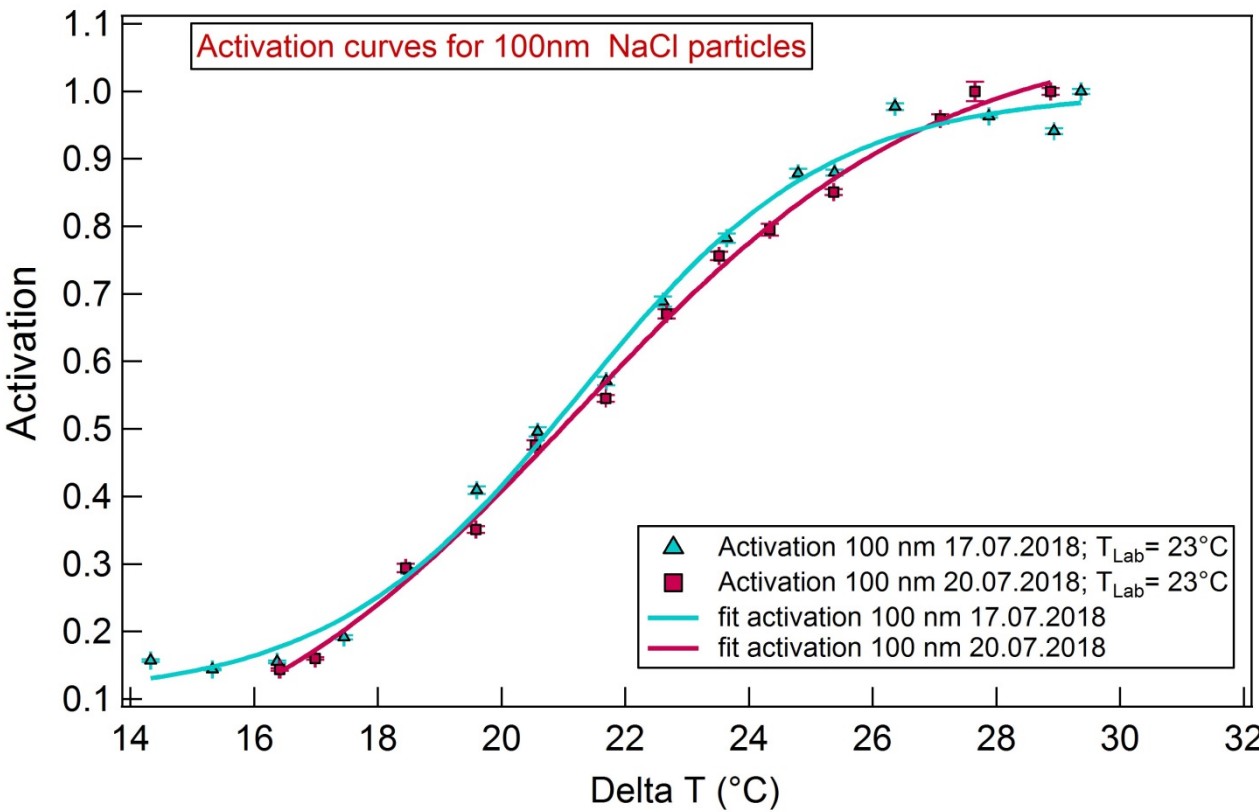

**Figure 4**: Activation curves for 100 nm NaCl particles measured on two different days. Both curves are equivalent.

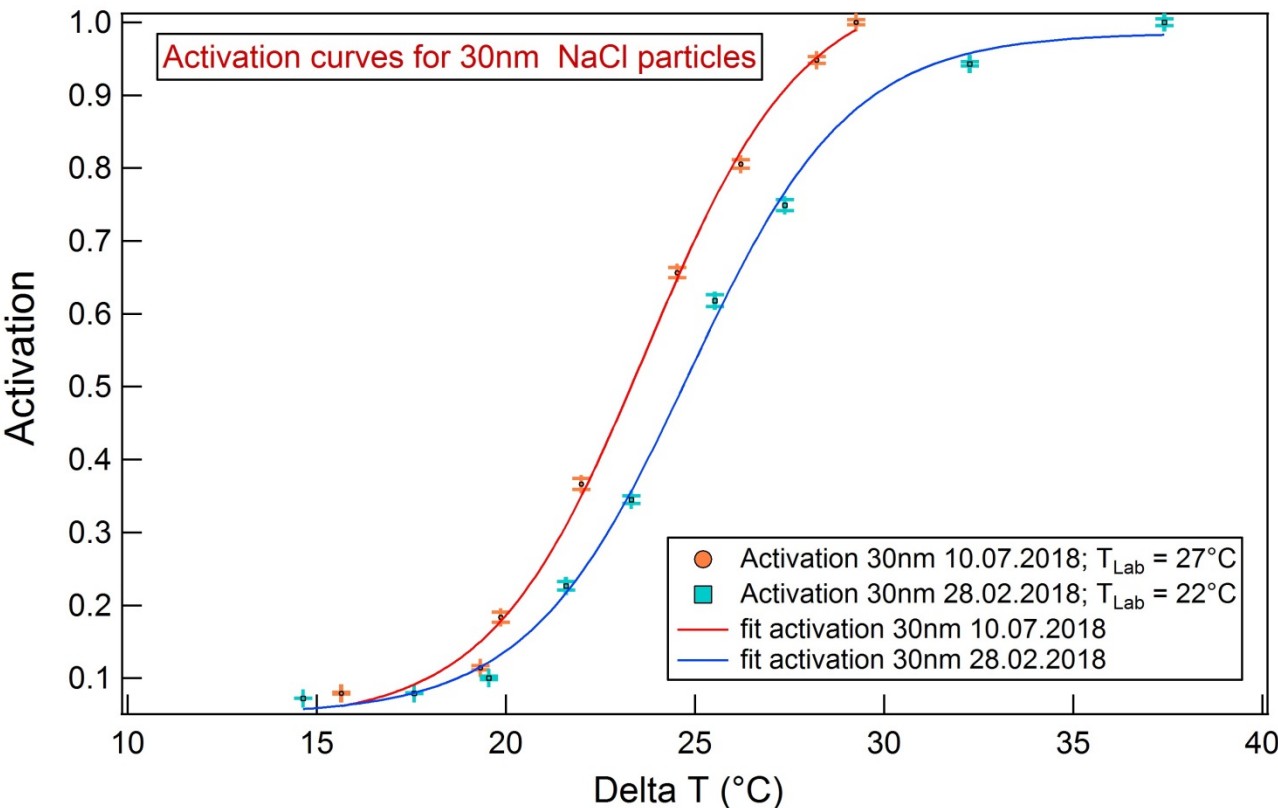

**Figure 5:** Activation curves for NaCl particles with 30 nm size measured in winter (blue curve, laboratory temperature 22 °C) and summer (red curve, laboratory temperature 27 °C).

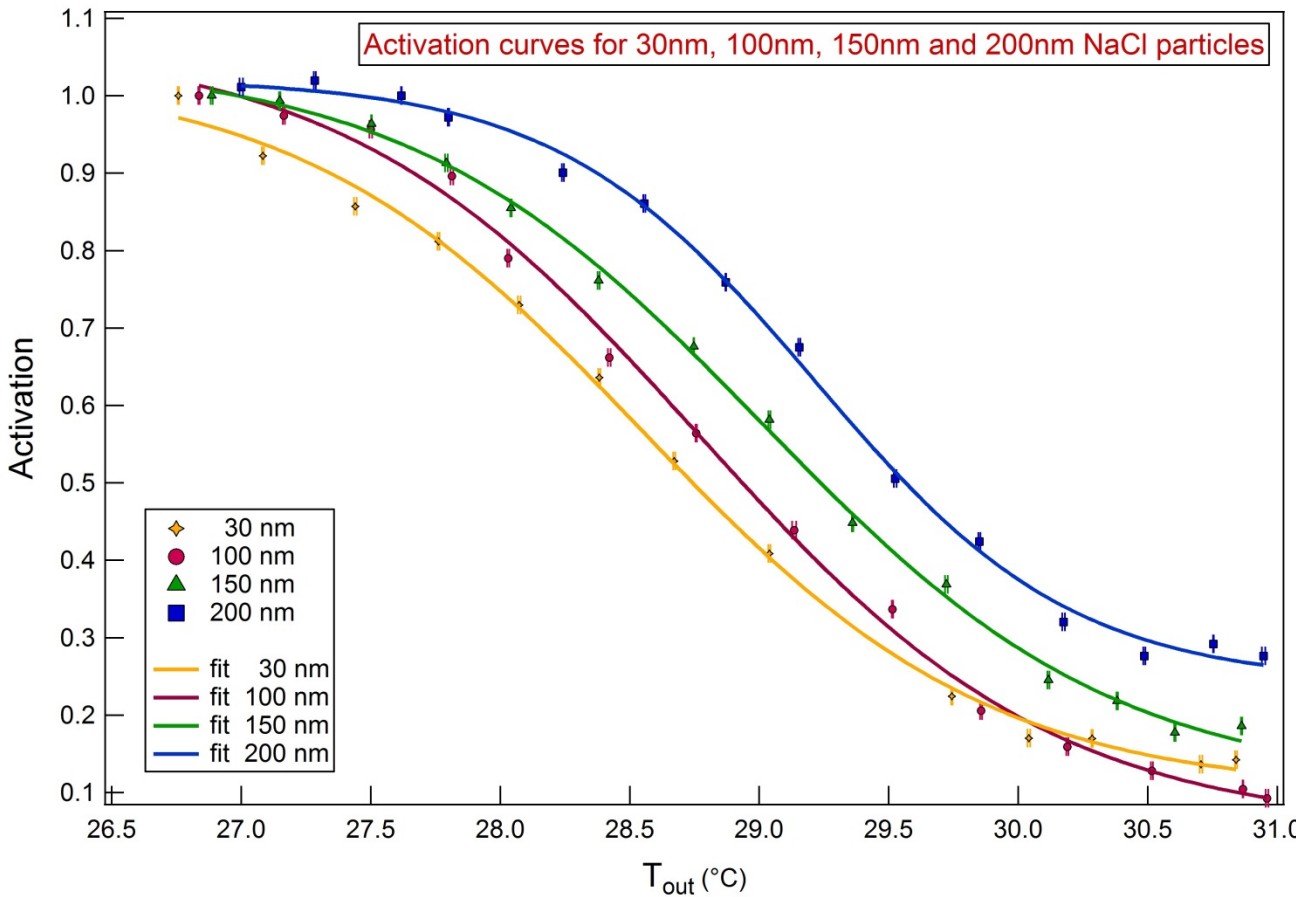

**Figure 6:** Activation curves as a function of $T_{out}$ for NaCl particles of different sizes. Measured points fitted with sigmoidal curves.

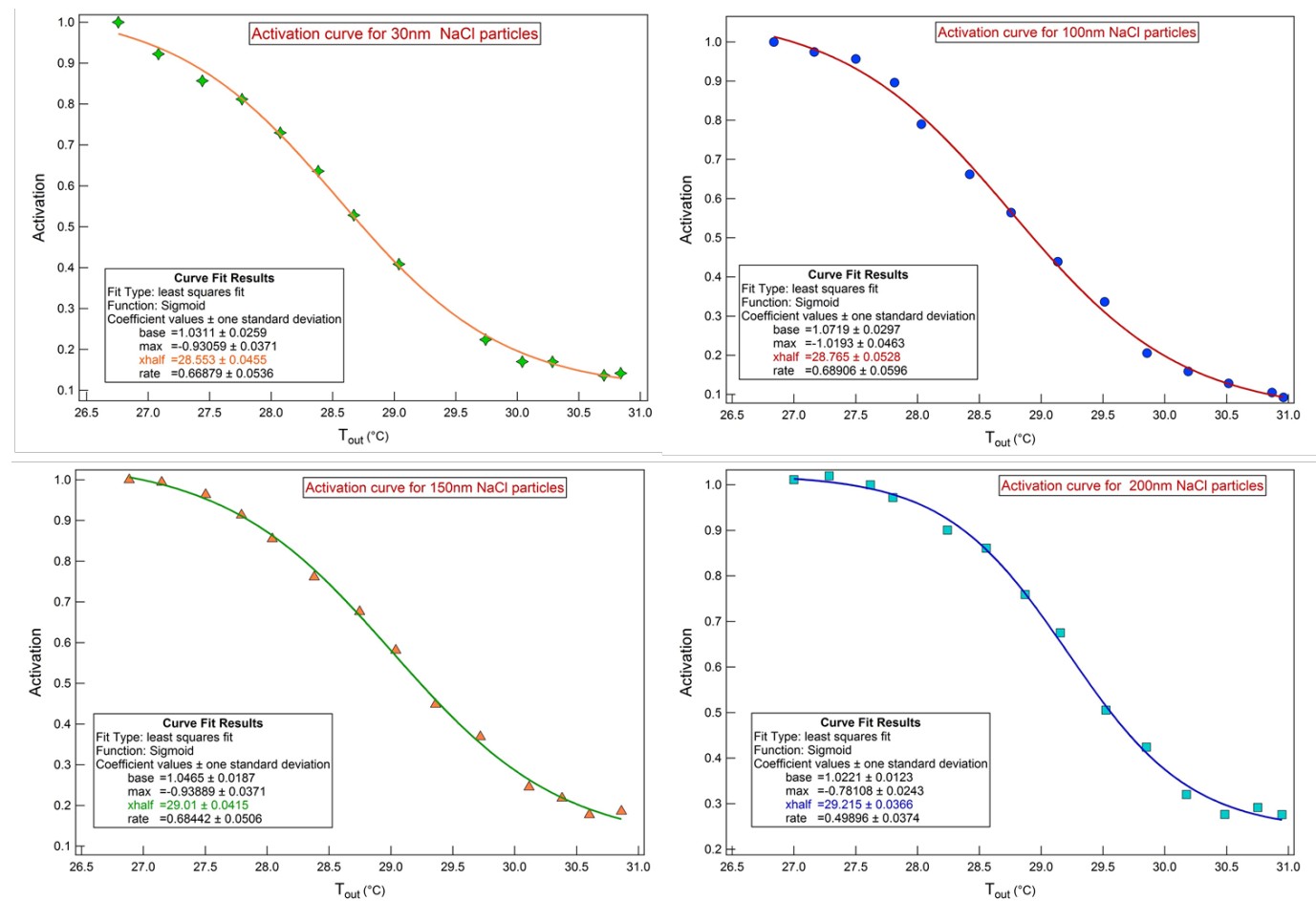

**Figure 7 a-d:** Determination of the critical supersaturation for 4 different activation curves.

| Parameters | Values |
|---|---|
| Van´t Hoff factor NaCl | 2 |
| Density, NaCl (kg/m³) | 2170 |
| Density, water (kg/m³) | 1000 |
| Surface tension of water at 20°C (N/m) | 0.072 |
| $M_w$ (kg/mol) | 0.018 |
| $M_s$ (kg/mol) | 0.059 |
| Universal gas constant (kJ/kmol K) | 8.314 |

**Table 1:** Parameters used to calculate the supersaturation for specific critical diameter and temperature (Weast, 1988).

| Particle size | Experimental concentration enrichment factor $EF_{exp}$ | Theoretical enrichment factor $EF_T$ | Enrichment efficiency (experimental/theoretical) |
|---|---|---|---|
| **NaCl** | | | |
| 30 nm | 16.6 | 21 | 79% |
| 100 nm | 16.4 | 21 | 78% |
| 150 nm | 17.0 | 21 | 81% |
| 200 nm | 16.8 | 21 | 80% |
| **PSL** | | | |
| 100 nm | 16.59 | 21 | 79% |
| 150 nm | 16.12 | 21 | 77% |

**Table 2:** Comparison between the measured experimental enrichment factors for NaCl and PSL particles with different sizes to the calculated theoretical enrichment factor. The measured enrichment factor corresponds to roughly 80% of the theoretical enrichment factor which agrees with the VACES efficiency given by Kim et al. (2001b). The measured enrichment factors do not depend on particle size and type.

| Particle size (nm) | $T_{out}$ for 50% activation (K) | $S_{crit}$ (%) |
|---|---|---|
| **NaCl** | | |
| 200 | 302.37 | 0.035 % |
| 150 | 302.16 | 0.054 % |
| 100 | 301.92 | 0.099 % |
| 30 | 301.71 | 0.602 % |
| **PSL** | | |
| 100 | 299.03 | 2.107 % |
| 150 | 300.35 | 1.394 % |

**Table 3:** Calculated critical supersaturations at the measured $T_{out}$ (at 50% activation) for NaCl and PSL particles of different sizes.

