# Peer review of "Versatile Aerosol Concentration Enrichment System (VACES) operating as a Cloud Condensation Nuclei (CCN) concentrator. Development and laboratory characterization."

_Atmospheric Measurement Techniques, 2019_

## Referee Comment (RC1) · Anonymous Referee #1 · 15 Apr 2019

The strength of the VACES system is its ability to enrich the sample, which allows for additional chemical analysis which is problematic in case of typically used CCN counters. However, based on the results presented I have some reservations how well it actually works. Although this is more like an introductory manuscript intended to demonstrate that VACES has some potential to operate as CCN counter, I would still like to see more evidence. Now the results are presented only for NaCl, which is a natural choice as it is well characterized material, but results with other particle types with lower hygroscopicity would be needed also. Based on the presented data it is

really difficult to estimate if the instrument accuracy is high enough for determining the CCN activity. Thus I recommend major revision before acceptance.

1) The instrument is described nicely, but more information could be added how long it actually takes to conduct a full scan of particle size and different temperatures. Also it is said that original VACES is modified to get into lower supersaturations, this could be elaborated more also.

2) There is always quite a high fraction of activated particles. Have the authors accounted for the multiple charging of particles in neutralizer? As the cut of size of virtual impactor is relatively low, part of that could be explained with multiple charged larger particles selected by DMA. However, activated fraction should still go closer to zero. The reason for high activated fraction should be explained.

3) Compared to actual differences in activation temperatures, the slope of activation curve is very gentle. Why is that? Do the particles experience highly different supersaturation conditions within the condenser? It is said that the flow is turbulent, but could this cause different growth behavior for different sized particles? Can the NaCl calibration be used also for particles with greatly lower hygroscopicity or should there some other component also? I see this as a big source of uncertainty and makes me wonder if it is possible at all to use the instrument outside laboratory conditions where the capability for chemical analysis would be really needed?

4) Is the data used for figures 3 and 6 the same. Behavior looks highly different and I doubt if observations are as reproducible as claimed by the authors. In Figure 3 the Delta T differs most for smaller particles whereas in Figure 6 T_out differs most for larger particles.

5) There is no variability (deviation from the mean) presented in the observations. Presenting that could give some information how stable the condenser is.

6) Measurement setup involves quite high temperature differences. This might cause

problems with particles of semivolatile nature like several organics or even nitrates. How do you account for this?

7) In Equation 2 the critical supersaturation is determined by the particles size. The temperature contribution is minimal and thus Table 3 gives no information which could somehow be used to evaluate the performance. Thus it could be removed.

---

## Referee Comment (RC2) · Anonymous Referee #2 · 21 May 2019

This is an experimental investigation manuscript that describes what the title offers— "..Development and laboratory characterization." It is an extension of previously used VACES technology to test operation with somewhat smaller sized particles than tested in the past.

In general, this paper does demonstrate that CCN can be concentrated fairly well within the constraints of artificially generated NaCl. Inclusion of particles other than NaCl would have added strength to the paper. Further, the current paper employed PM over size range of 30 nm to 200 nm. It would be useful to provide discussions on the typical

profile of CCN characteristics and justify the choice of experimental conditions. The current manuscript needs minor revision before acceptance for publication.

Specific comments Section 4.1 NaCl is used for performance testing and the text indicates that there were no changes in chemistry or physical nature of collected in PM with VACES. There are very limited discussions to cover this point. How might other hygroscopic aerosols behave in terms of size/shape impacts of growth and drying process?

Section 4.2 "The order of the activation curves agrees with theory". This statement needs expansion. Further, figures seem to show quite a difference in efficiency of performance between small and larger test PM. Do the authors suggest that corrections can be made to adjust these values to account for observed differences? It also seems that at particles smaller than 30nm this could be an even larger issue. Further discussions on implications are suggested to be included.

Section 4.3 "Further results also demonstrate that the activation curves do not depend on the inlet concentration." The concentrations of NaCl had a maximal count of 6000 /cm3— (monodisperse at 100nm). It would be useful to expand this statement to deal with what might be expected in terms of performance in real world.

Conclusions can be expanded to provide recommendations for future application in real world operations and extend beyond proofs by NaCl.

The closing sentences need a bit of cautionary text related to the real world applications—especially with regards to temperature, size, chemistry issues. "Notwithstanding the strong temperature dependence, we found that the CCN VACES is a reliable instrument to activate CCN and enrich CCN concentrations at low supersaturations, provided that the temperature settings are carefully controlled."

---

## Author Comment (AC1) · 17 Jun 2019

**Authors´response to the Referee 1 comments on manuscript titled** *" Versatile Aerosol Concentration Enrichment System (VACES) operating as a Cloud Condensation Nuclei (CCN) concentrator. Development and laboratory characterization."* **submitted to AMT 25th February 2019.**

The authors would like to thank the reviewer for the helpful comments. Our responses are given in blue after each of the comments (bold characters). In the revised MS, changes concerning the comments of the reviewers are indicated in red. For a better understanding the representation of the virtual impactor in Fig. (1) was modified. Small editorial changes (such as corrections of typos, use of symbols instead of full text, etc.) are not reported here, as they are irrelevant for the scientific content of the paper.

**However, based on the results presented I have some reservations how well it actually works. Although this is more like an introductory manuscript intended to demonstrate that VACES has some potential to operate as CCN counter, I would still like to see more evidence. Now the results are presented only for NaCl, which is a natural choice as it is well characterized material, but results with other particle types with lower hygroscopicity would be needed also.**

In the revised MS, results from activation curves of 100 nm and 150 nm Polystyrene Latex particles (PSL) which have a less hygroscopic behaviour are added (see Table 2 and 3; page 20). The maximum experimental enrichment factor obtained from the PSL measurements is over 16 which is very similar to the experimental enrichment factor obtained for NaCl particles. The experimental enrichment factor for 100 nm PSL particles at different ΔT is incorporated in Fig. 2 (page 16) and shows the correct performance of the new CCN-VACES performance also for less hygroscopic particles. Some recent publications were referenced that provide a detailed description of VACES operation and demonstrate its potential as particle concentrator (page 10).

Zhi N., Moore, K.F., Polidori, A., and Sioutas, C. "Field Validation of the new miniature Versatile Aerosol Concentration Enrichment System (mVACES)". Aerosol Science and Technology, 40 (12), 1098-1110, 2006

Saarikoski S., Carbone S, Cubison MJ, Hillamo R, Keronen P., Sioutas C., Worsnop D.R, Jimenez J.L. "Evaluation of the performance of a particle concentrator for on-line instrumentation". Atmospheric Measurement Technologies, 7, 2121-2135, 2014

Zhao, Y., Bein, K.J., Wexler, A.S., Misra, C., Fine, P.M. and C. Sioutas, C. "Using a Particle Concentrator to Increase the Hit Rates of Single Particle Mass Spectrometers". Journal of Geophysical Research, 110 (D7): Art. No. D07S02, 2005

**Based on the presented data it is really difficult to estimate if the instrument accuracy is high enough for determining the CCN activity.**

The activations curve as well as their reproducibility (under stable ambient conditions) are a clear evidence of the proper functioning of the new CCN-VACES. We do suggest, however, that the system should be calibrated by recording an activation curve with test aerosol particles previous to a measurement series. If the modified VACES had not functioned properly, the enrichment factors would not have been comparable to studies (Kim et al., 2001a).

**1) The instrument is described nicely, but more information could be added how long it actually takes to conduct a full scan of particle size and different temperatures. Also it is said that original VACES is modified to get into lower supersaturations, this could be elaborated more also.**

A more detailed paragraph has been incorporated in the manuscript. Modifications from the original VACES are explained in (page 4, lines 19-25). More information on the measurement procedure has also been added (page 7, lines 3-5; page 7, lines 15-23).

**2) There is always quite a high fraction of activated particles. Have the authors accounted for the multiple charging of particles in neutralizer? As the cut of size of virtual impactor is relatively low, part of that could be explained with multiple charged larger particles selected by DMA. However, activated fraction should still go closer to zero. The reason for high activated fraction should be explained.**

Doubly and higher charged particles may have been the reason why the activation curves do not go closer to zero. An indication for this effect is the apparent size dependence of the start of the curves – larger particles will have more multiple charges. According to the study by Rose et al., (2008) the influence of doubly charged particles appears in the activation curves as first plateau in their activation curves plotted as activation ratio vs. particle size at constant supersaturation. Similar curves were obtained in an earlier (unpublished) study with our static thermal diffusion chamber CCNC (Vera Meyer, Kondensationskernaktivierung des wasserlöslichen Anteils in urbanem Aerosol, Diploma Thesis, University of Vienna, 2006; see figure below). In our case, the activation curves are plotted as functions of delta T (or T_out) for constant particle size. The 50% point of these curves, however, which is used to calculate supersaturations, is not influenced by this effect.

[Figure]

V. Meyer, Diploma Thesis, University of Vienna

**3) Compared to actual differences in activation temperatures, the slope of activation curve is very gentle. Why is that?**

The slope depends of course on the spread of the x-axis. We chose a large spread, as the difference in T_out is so small for curves obtained for particles of different diameters. In general, the slope of activation curves varies depending on particle size and / or supersaturation. The activation curves obtained e.g. by Giebl et al. (2002) (activation ratio plotted vs. dry diameter) are steeper at higher supersaturations (SS = 1.08 % and SS = 0.42%) than at lower ones.

**Do the particles experience highly different supersaturation conditions within the condenser?**

In the condenser tube the turbulent flow regime ensures a radially quite homogeneous mixture of water vapour. Supersaturation increases along the tube axis.

**It is said that the flow is turbulent, but could this cause different growth behavior for different sized particles?**

The growth behaviour of a particle depends on its activation according to Köhler theory, so of course differently sized particles (if present; we used monodisperse particles) will grow differently at specific water vapour supply and temperature. Of course some droplets might be lost to the walls in the turbulent flow, which might be a reason for the difference between theoretical and experimental enrichment factor.

**Can the NaCl calibration be used also for particles with greatly lower hygroscopicity or should there some other component also? I see this as a big source of uncertainty and makes me wonder if it is possible at all to use the instrument outside laboratory conditions where the capability for chemical analysis would be really needed?**

We used NaCl to calibrate the instrument, i.e. to see which T_out gives which supersaturation (at constant T_s), so NaCl can be seen here as a "supersaturation sensor". The new data obtained with PSL show that PSL particles are also activated at the expected supersaturation (obtained from Kelvin theory). The supersaturation achieved in the CCN-VACES depends on the temperature settings, and not on the chemical nature of the particles. Once the instrument's temperature settings are well established and stabilized, we can determine the supersaturation in the system and enrich other different aerosols particles. Complex aerosols containing volatile or semi-volatile material, however, might experience some changes in the temperature range used in the system. This is true, however, also for other CCN counters that operate at elevated temperatures.

**4) Is the data used for figures 3 and 6 the same. Behavior looks highly different and I doubt if observations are as reproducible as claimed by the authors. In Figure 3 the Delta T differs most for smaller particles whereas in Figure 6 T_out differs most for larger particles.**

The observation is correct, but the figures are correct, too. We clarified this in the revised MS. As the reviewer pointed out, activation curves in Fig. (3) for smaller particles correspond to higher $\Delta T$ values while the activation curves for larger particles correspond to lower $\Delta T$ values. The activation curve for 30 nm particles is shifted to the right, i.e. to higher $\Delta T$, which is the temperature difference between the saturator temperature and the condenser temperature ($T_s$-$T_c$). If the saturator temperature is fixed and remains constant, a higher $\Delta T$ means lower condenser temperatures $T_c$ . As seen in Fig. (3), 200 nm particles are activated at lower $\Delta T$ which implies higher $T_c$ and consequently a lower critical supersaturation.

A more detailed paragraph has been incorporated in the manuscript (page 8, lines 24-29). In Fig. (6), activation curves are plotted vs. T_out. High T_out correspond to low supersaturations, so the curves for small particles needing higher supersaturations (and lower T_out) to activate are shifted to the left.

An extended explanation has been incorporated in the manuscript (page 9, lines 29-30) and page 10, lines 2-4).

**5) There is no variability (deviation from the mean) presented in the observations. Presenting that could give some information how stable the condenser is.**

As the temperature difference between the curves is relative small errors bars were omitted in order to show the curves more clearly.

**6) Measurement setup involves quite high temperature differences. This might cause problems with particles of semivolatile nature like several organics or even nitrates. How do you account for this?**

Previous studies (Sioutas et al., 1998; Saarikosi et al., 2014) performed with the original VACES have demonstrate that the concentration enrichment does not depend on the composition of the measured aerosol. The experimental concentration enrichment factor obtained for (hygroscopic) ammonium sulfate, (semi-volatile) ammonium nitrate, non-hygroscopic PSL and indoor air was practically the same. As in many other instruments, particles containing volatile compounds may be changed, but the VACES not designed to characterize volatile species.

**7) In Equation 2 the critical supersaturation is determined by the particles size. The temperature contribution is minimal and thus Table 3 gives no information which could somehow be used to evaluate the performance. Thus it could be removed**.

We decided to keep Table 3, as it gives the temperature settings needed to set supersaturations in cases where the chemical composition of the aerosol is unknown.

---

## Author Comment (AC2) · 17 Jun 2019

Authors´response to the Referee 2 comments on manuscript titled " *Versatile Aerosol Concentration Enrichment System (VACES) operating as a Cloud Condensation Nuclei (CCN) concentrator. Development and laboratory characterization.*" submitted to AMT 25[th] February 2019.

The authors would like to thank the reviewer for the helpful comments. Our responses are given in blue after each of the comments (bold characters). For a better understanding the representation of the virtual impactor in Fig. (1) was modified. In the revised MS, changes concerning the comments of the reviewers are indicated in red. Small editorial changes (such as corrections of typos, use of symbols instead of full text, etc.) are not reported here, as they are irrelevant for the scientific content of the paper.

**In general, this paper does demonstrate that CCN can be concentrated fairly well within the constraints of artificially generated NaCl. Inclusion of particles other than NaCl would have added strength to the paper.**

In the revised MS, results from activation curves of 100 nm and 150 nm Polystyrene Latex particles (PSL) which have a less hygroscopic behaviour are added (see Table 2 and 3; page 20). The maximum experimental enrichment factor obtained from the PSL measurements is over 16 which is very similar to the experimental enrichment factor obtained for NaCl particles. The experimental enrichment factor for 100 nm PSL particles at different ΔT is incorporated in Fig. (2) (page 16) and shows the correct performance of the new CCN-VACES performance also for less hygroscopic particles. Some recent publications were referenced that provide a detailed description of VACES operation and demonstrate its potential as particle concentrator (page 10) .

**Further, the current paper employed PM over size range of 30 nm to 200 nm. It would be useful to provide discussions on the typical profile of CCN characteristics and justify the choice of experimental conditions.**

More detailed characteristics of CCN have been incorporated in the manuscript (page 3, lines 14-17). The VACES modifications were performed in order to get down to the low supersaturations commonly occurring in the atmosphere. McFiggans et al. (2006) point out the complexity of ambient aerosol composition, which is relevant for CCN activation in a limited size range from ca. 30 nm up to 200 nm. Particles smaller than 30 nm are unlikely to activate and particles larger than 200 nm will activate anyway, as they normally contain enough soluble material.

**Section 4.1 NaCl is used for performance testing and the text indicates that there were no changes in chemistry or physical nature of collected in PM with VACES. There are very limited discussions to cover this point.**

We intentionally used NaCl for the calibration measurements, as it is a well characterized non-volatile substance. Volatile or semi-volatile material might undergo changes in the temperature range used here. A sentence to this effect was put into the MS in the introduction and also in the conclusion section. A detailed discussion of these effects, however would be far beyond the scope of this MS.

**How might other hygroscopic aerosols behave in terms of size/shape impacts of growth and drying process?**

Other hygroscopic aerosols would also activate according to Koehler theory at their individual critical supersaturations. Particle shape might have an influence, but Koehler theory posits spherical particles, which is usually a valid assumption, as hygroscopic particles deliquesce already at relative humidities below 100% and are spherical by the time they activate.

**Section 4.2 "The order of the activation curves agrees with theory". This statement needs expansion.**

An extended explanation has been incorporated in the manuscript (page 8, lines 24-29).

**Further, figures seem to show quite a difference in efficiency of performance between small and larger test PM. Do the authors suggest that corrections can be made to adjust these values to account for observed differences? It also seems that at particles smaller than 30nm this could be an even larger issue. Further discussions on implications are suggested to be included.**

The differences in the steepness of the slopes of the activation curves for small and larger particles is a usual feature found in many studies. As presented e.g.by Giebl et al. (2002), the activation curves (activation ratio plotted vs. dry diameter) are steeper at higher supersaturations (SS = 1.08 % and SS = 0.42%) than at lower ones. Activation curves obtained from urban aerosol (Burkart et al., 2012) show the same behaviour. An extended explanation has been incorporated in the manuscript (page 9, lines 29-30).

**Section 4.3 "Further results also demonstrate that the activation curves do not depend on the inlet concentration." The concentrations of NaCl had a maximal count of 6000 /cm3 (monodisperse at 100nm). It would be useful to expand this statement to deal with what might be expected in terms of performance in real world.**

In the urban background aerosol of Vienna, typical total CN concentrations lie between 1000 - 8000 #/cm³ which corresponds to the concentrations used for the calibration, so this is the local "real world". Nevertheless higher particle number concentrations may occur in heavily polluted regions. The limiting factor here might be the available water vapour in the condenser tube(s). Earlier studies given in the experiment section, however, have not seen an effect of aerosol concentration on the enrichment factor. For example, Ntziachristos et al. (2007) used the VACES in the vicinity of the busiest US freeway in diesel truck traffic, with particle concentrations in the 30000-500000 #/cm$^3$ range and saw no effect in the performance of the system regarding the enrichment factor.

Ntziachristos L, Zhi N., Geller M.D., Sheesley R., Schauer J.J. and Sioutas C. "Fine, Ultrafine and Nanoparticles Trace Element and Metal Composition Near a Freeway With Heavy Duty Diesel Traffic." *Atmospheric Environment,* 41 (27): 5684 – 5696, 2007.

**Conclusions can be expanded to provide recommendations for future application in real world operations and extend beyond proofs by NaCl.**

As we have shown in the manuscript NaCl is a good aerosol to calibrate the CCN-VACES in terms of supersaturation. Once the CCN-VACES is set to a certain supersaturation, it will enrich CCN for further chemical analysis. A caveat has been added to the conclusions as to the possible changes of volatile particles in the temperature range of the VACES.

**The closing sentences need a bit of cautionary text related to the real world applications. Especially with regards to temperature, size, chemistry issues. "Notwithstanding the strong temperature dependence, we found that the CCN VACES is a reliable instrument to activate CCN and enrich CCN concentrations at low supersaturations, provided that the temperature settings are carefully controlled."**

See above.

---

## Author Response (AR2)

**Authors´response to the Referee 3 and the Editor comment on manuscript titled "** *Versatile Aerosol Concentration Enrichment System (VACES) operating as a Cloud Condensation Nuclei (CCN) concentrator. Development and laboratory characterization."* **submitted to AMT 25ᵗʰ February 2019.**

Our response is given in blue after the comment, which is shown in bold characters. In the revised MS, changes concerning the reviewer's comment are indicated in red. The colour code in Figure 4 was corrected (the colours of the data points and the fitted curves had been the wrong way round).

**I agree with the Authors with most of the replies, but disagree in one certain point. The lack of "error bars" in different figures was raised up, and I'm not happy with the answer "As the temperature difference between the curves is relative small errors bars were omitted in order to show the curves more clearly." Especially in the case when the differences are small the uncertainty should be presented in a way or another. Without any statistics presented the conclusions made on the observations can be too optimistic. I'm not asking it to be added to every single plot, but at least one with some discussion.**

In the revised MS error bars were added to Figures 3 – 6. As each data point actually corresponds to ca. 300 individual measurement points, the error bars correspond to the standard error of the mean.